# Parental Perception of the Oral Health-Related Quality of Life of Children and Adolescents with Autism Spectrum Disorder (ASD)

**DOI:** 10.3390/ijerph20021151

**Published:** 2023-01-09

**Authors:** Anna Cecília Farias da Silva, Taís de Souza Barbosa, Maria Beatriz Duarte Gavião

**Affiliations:** 1Department of Health Sciences and Pediatric Dentistry, Piracicaba Dental School, University of Campinas, Piracicaba 13414-903, SP, Brazil; 2Department of Dentistry, Life Sciences Institute, Federal University of Juiz de Fora, Governador Valadares 35020-360, MG, Brazil

**Keywords:** autism spectrum disorder, oral health, quality of life, children, adolescents

## Abstract

This study evaluated the parental perception of the oral health-related quality of life (OHRQoL) of children and adolescents with autism spectrum disorder (ASD) and their family functioning. Moreover, sociodemographic factors associated with parental ratings of OHRQoL were assessed. A hundred parents/guardians of children and adolescents aged 6 to 14 years with ASD (ASD group) and 101 unaffected children and adolescents (UCA group) participated. Data collection was carried out using a Google form, containing three sections: (1st) Socioeconomic data and health history; (2nd) Oral health assessment by parental report; (3rd) The short forms of the Parental-Caregiver Perceptions Questionnaire (16-P-CPQ) and the Family Impact Scale (4-FIS). The scores of 16-P-CPQ total and subscales and 4-FIS were significantly higher for the ASD group (*p* < 0.02), except for the oral symptoms subscale (*p* > 0.05). Older ages (OR = 1.24), brushing 0/1x day (OR = 2.21), teeth grinding (OR = 2.20), gingival bleeding (OR = 3.34), parents with an elementary school degree (OR = 0.314) and family incomes less or equal to the minimum wage (OR = 3.049) were associated with a worse OHRQoL. Parents in the ASD group had a worse perception of QHRQoL when compared to the UCA group. ‘Frequency of tooth brushing’, ‘gingival bleeding’, and ‘teeth grinding’ were predictors of the worst parental perception of their children’s OHRQoL. Families with low socioeconomic conditions were more strongly affected by the oral conditions of their children.

## 1. Introduction

Autism Spectrum Disorder (ASD) is a neuropsychiatric condition characterized by multiple delays and behavioral deviations that manifest during early childhood [1]. Early diagnosis is extremely important, as it allows a multidisciplinary team and the parents to devise treatment strategies to stimulate skills in these children [2]. Children with ASD have limitations that make it difficult to maintain their general health and, consequently, oral health. This makes them dependent on help from family members and caregivers to carry out daily activities, such as oral hygiene [3,4]. Some studies have shown that children with ASD are part of an at-risk group, as they are more prone to the development of caries lesions, periodontal diseases, and traumatic injuries [5,6]. Other oral concerns in ASD have been considered, such as tooth loss, malocclusion, bruxism and tooth grinding [7]. In a recent review, it was noted that ASD subjects present a higher risk for increased overjet, but not for other types of malocclusions [8]. The evidence of bruxism and grinding teeth/clenching being more likely in individuals with ASD is very low, due to the high heterogeneity and high risk of bias in primary studies [9,10].

It is not just the presence or absence of a disease/condition that matters, but how it affects an individual’s daily life or quality of life [11]. Oral health-related quality of life (OHRQoL) is a multidimensional construct, based on the subjective assessment of oral health, including the analysis of the patients functional and emotional well-being, expectations, and satisfaction with health care. The assessment of quality of life allows for a change from traditional health dynamics in which only the cure of the disease is achieved, to focusing on the patient’s social and emotional experience, treating the patient as a whole [12]. OHRQoL is a good indicator of the efficiency of health interventions, and it also facilitates the establishment of at-risk groups and optimal health guidelines [12].

Some tools have been developed over the years to assess the impact of oral health on children’s and adolescents’ quality of life, for example, the *Child Perceptions Questionnaires* (CPQ_8–10_ and CPQ_11–14_) answered by children [11,13]. In some situations, it is not possible to access health information through the child, because of their age or disability, so it is necessary that parents inform the OHRQoL assessment, using appropriate questionnaires, such as the *Parental–Caregiver Perceptions Questionnaire* (P-CPQ) and the *Family Impact Scale* (FIS). A number of these OHRQoL scales have been translated and validated worldwide, including the Brazilian Portuguese version [14,15,16].

Oral diseases can also affect the school routine and leisure activities, as well as self-esteem and social relationships. The psychosocial impact of these diseases often reduces children’s quality of life [17].

In addition, inequalities in the access to healthcare services, especially for individuals with disabilities, such as the high cost of treatments, transportation difficulties, lack of knowledge about oral health, non-cooperative behavior of the child all negatively impact the child’s quality of life [18,19]. Furthermore, the behavioral difficulties of children and adolescents with ASD decrease their collaboration with oral hygiene, even if performed by their parents, making them prone to severe caries experiences [19,20] and periodontal diseases [5,21]. Some studies have revealed a negative impact of high caries prevalence on OHRQoL [3,19,20]. Children and adolescents with ASD appear to have a poorer quality of life compared to the unaffected population [22,23,24]. On the other hand, other studies have revealed that, according to the parents’ perception, dental treatment has a positive impact on the OHRQoL of children and adolescents with ASD [25,26,27].

Assessing the impact of oral diseases on the quality of life is important for planning and promoting the general health of children, especially those with ASD. Parental perception is a great tool for this assessment, in addition to motivating them to access oral healthcare services periodically.

This study aimed to evaluate the parental perception of oral health-related quality of life (OHRQoL) of children and adolescents with autism spectrum disorder (ASD) and their family functioning and identify sociodemographic factors that are associated with parental ratings of OHRQoL in a Brazilian population of children and adolescents.

## 2. Materials and Methods

This observational cross-sectional national study was carried out online, including a sample composed of parents/guardians of children and adolescents with ASD and an unaffected population (UCA group). The study was carried out from October 2021 to July 2022. The study was conducted according to the guidelines of the Declaration of Helsinki and approved by the ethics committee of Piracicaba Dental School, University of Campinas, Piracicaba, SP, Brazil (protocol code: 51610921.2.0000.5418–20 October 2021). Informed consent was obtained from all subjects involved in the study.

The sample size was calculated based on the short form of the P-CPQ total score [28] reported in a longitudinal study [27] in Brazilian children and adolescents with ASD, aged 6 to 14 years. Considering, at baseline, a mean total P-CPQ score of 13.2, a standard deviation of 6.4, a sampling error of 10%, and a confidence level of 95%, the required sample size was defined as 91 individuals with ASD.

Initially, Brazilian multidisciplinary institutions that work with applied behavior analysis (ABA) and elementary schools in Brazil were searched on the Google website. After this, the institutions with available e-mails in their website or Instagram page were contacted. An invitation letter was sent to 168 selected institutions and schools, clarifying the objectives and methodology of the study, and requesting help in recruiting parents/guardians. Of these, 53 agreed to help recruit and shared the link to the Google forms with their social media parent groups (Figure 1). These forms contained information about the study, the informed consent forms for participants to participate, and forms for data collection. The response rate of the institutions contacted was 31.5%.

Inclusion criteria were parents of children and adolescents aged 6 to 14, of either sex, residing in different states of Brazil, who were previously diagnosed with ASD by a child psychiatrist or pediatric neurologist and classified according to the degree of autism. They have been assisted in multidisciplinary centers. Moreover, parents of unaffected schoolchildren were also included. The exclusion criteria were parents/caregivers of children and adolescents who, at the time of recruitment, had other systemic disorders, such as neurological disorders, cerebral palsy, and other chronic diseases that could interfere with oral health, and parents with a cognitive problem that could compromise answers to the questionnaires, or were unable to answer the online forms (no access to social network or internet).

Data collection was carried out by completing a Google form, which was divided into three sections: the first contained a socioeconomic family questionnaire (parents’ education, family income, employment status) and the child’s or adolescent’s health history (age at diagnosis and ASD degree, use of medicines, health problems); the second addressed the assessment of oral health according to the parental report (frequency of tooth brushing and sugar consumption, deleterious oral habits); in the third, two OHRQoL questionnaires were applied, the Brazilian Portuguese short forms of the *Parental–Caregiver Perceptions Questionnaire* (P-CPQ) and the *Family Impact Scale* (FIS).

The P-CPQ is a self-administered instrument, which aims to assess parental perceptions of their children’s or adolescents’ OHRQoL. The short versions, developed from the most impacted items on the Brazilian Portuguese P-CPQ and FIS [14,15,16], consisted of 16 and 4 items, respectively (Appendix A). 16-P-CPQ is subdivided into four subscales: oral symptoms (OS), functional limitations (FL), emotional well-being (EWB), and social well-being (SWB); plus, two global questions about the parental overall perception of the child’s oral health (OH) and how much the oral or orofacial condition affects the child’s overall well-being (OWB). 4-FIS is a scale to evaluate the effects of a child’s oral conditions on family functioning and activities.

Both questionnaires assess the frequency of oral events in the three months prior to their application from the perspective of parents or caregivers. 16-P-CPQ and 4-FIS responses are expressed on a five-point Likert scale with scores ranging from 0 “Never” to 4 “Every day or almost every day”; answers such as “Don’t know” were allowed and were assigned the “0” score. OH and OWB ratings, using a five-point Likert scale, range from “Excellent” to “Poor” and from “Not at all” to “Very much”, respectively. At the end, the scores are computed by summing the items of each subscale and all items for total scale, for FIS, and for global perceptions. The total 16-P-CPQ and 4-FIS scores range from 0 to 64 points and 0 to 16 points, respectively. Moreover, the global perceptions range from 0 to 8 points; this score summarizes the caregiver’s perception of the child’s OHRQoL, using two questions (OH and OWB) and also tests for construct validity. A high final score indicates a worse OHRQoL.

To assess test–retest reliability, 20 participants (10 from each group) were randomly selected to answer the questionnaires a second time, a month later.

### Statistical Analysis

The collected data were analyzed using the Jamovi software version 2.3.18 (The Jamovi project, 2021, retrieved from https://www.jamovi.org (accessed on 27 September 2022). To verify the distribution of data, the Shapiro–Wilk test was applied. Descriptive statistics consisted of calculating frequencies, mean, standard deviation, median, and interquartile ranges. The chi-square test was used to assess the proportion of variables. A Mann–Whitney non-parametric statistical test was used to compare questionnaire data between groups and effect size by ranking biserial correlation, since the scores were not normally distributed. The internal consistency of the questionnaire was evaluated by Mcdonald’s Omega, and the intraclass correlation coefficient was used to assess the test–retest reliability, calculated by two-way mixed effects. The strength of agreement between the scores was based on the following standards for intraclass correlation coefficient (ICC): <0.2, poor; 0.21–0.40, fair; 0.41–0.60, moderate; 0.61–0.80, substantial; and 0.81–1.0, excellent to perfect [29]. The ICC was calculated for the total scale, subscale, global perception, and FIS. To test the validity of parental perception of the child’s OHRQoL, Spearman’s correlation between the global perception scores and the 16-P-CPQ and 4-FIS scores was applied. In addition, logistic regression models were constructed to verify potential associations between the variables, considering the 16-P-CPQ total and 4-FIS as the dependent variables and the respective median values were used as the threshold for the outcomes. 16-P-CPQ values above the median represented a poorer parental perception of the child’s OHRQoL and 4-FIS represented a poorer family functioning and activities due to the oral conditions of the child. First, a bivariate logistic regression was set and variables with associations with a *p* value equal or less than 0.15 were entered into the multivariate model using hierarchical entry procedures. The alpha value was set at 95%.

## 3. Results

Figure 1 summarizes the steps for obtaining the sample. The response rate of the institutions contacted was 31.5%.

A total of 201 parents were recruited, forming two groups: the ASD group, parents of children and adolescents with ASD (n = 100) and the UCA group, parents of unaffected children and adolescents (n = 101). Although data were collected using self-completed questionnaires, there were no missing data, due to the characteristics of the online forms, which did not allow for leaving any question unanswered.

The description of the sociodemographic variables, oral habits, and dental care of both groups of parents is presented in Table 1. The age of the UCA group was higher than the ASD group. However, the number of boys was similar to girls, whereas the number of boys in the ASD group was greater than girls. There was a predominance of ASD diagnosis between 2–6 years (*p* < 0.001) and level 2, moderate, was the most frequent diagnosis (*p* < 0.001). As expected, a higher number of children and adolescents with ASD were routinely taking medication (*p* < 0.01). The most reported medications were risperidone, Ritalin and associated formulations. Family characteristics was similar for the two groups.

Despite the age, six years or older, a higher frequency of bottle-feeding was observed in the ASD group. However, most children and adolescents in the two groups no longer used it. Having a regular mealtime was predominant, and both groups consumed sugar, without a difference in the number of times per day between them. Half of the children and adolescents of the ASD group did not brush or only brushed their teeth once a day. The other half brushed two to three times a day, mostly in the UCP group, differing significantly (*p* < 0.001). Furthermore, 75% of the parents of the ASD group were responsible for oral hygiene (*p* < 0.001), probably due to the characteristics of ASD.

Although a high percentage of all the children and adolescents visited a dentist periodically, the frequency of the UCA group was higher (*p* < 0.001). Premature tooth loss was more frequent in the ASD group (*p* < 0.001), whereas no association between groups was observed for gingival bleeding or trauma. It is noteworthy that the oral characteristics were reported by the parents through the forms, but most of the respondents were mothers, ensuring credibility. Regarding deleterious oral habits, only the use of bottle-feeding was more frequent in the ASD group, corroborating the above comments. Moreover, mouth breathing and teeth grinding showed a significant association with ASD, since the respective proportions were higher for the ASD group.

Table 2 shows the descriptive data of the questionnaires, considering the 16-P-CPQ total, their subscales, and 4-FIS by groups and the respective comparisons. The scores of the 16-P-CPQ total scale and subscales were significantly higher for the ASD group (*p* < 0.02) except for the OS subscale (*p* = 0.096). Global perception also presented higher scores for children and adolescents with ASD (*p* < 0.001). These differences revealed a worse parental perception of those children’s oral health and OHRQoL. Nevertheless, the effect sizes were of low magnitude. 4-FIS scores showed that families of the ASD group seemed to be more affected by their child’s oral health status and their need for treatment seemed to have a stronger impact on the family’s routine (*p* = 0.025).

The descriptive data of the global perception questions (OH and OWB) are presented in Table 3. Among the parents of the ASD group, 38% considered their child’s oral health as fair/poor, while in the other group these ratings were only 6%. Regarding the overall well-being, 24% of the parents of the ASD group considered their child to be somewhat/very affected by oral health/condition.

In both groups (Table 4), the scale reliability statistics showed a McDonald’s omega for the 16-P-CPQ total scale greater than 0.7, indicating a substantial internal consistency reliability. The OS and FL subscales presented a McDonald’s omega from 0.502 to 0.692, determining a moderate internal consistency reliability, whereas the respective values for the EWB and SWB subscale were less than 0.5, meaning a weak internal consistency reliability. The item reliability statistics showed no substantial increase in the McDonald’s omega if an item was excluded.

The test–retest performed with 20 parents (10 from each group) after one month of the first application, showed a positive result (Table 4). The ICC was 0.982 for the 16-P-CPQ and 4-FIS and ranged from 0.969 to 1 in the subscales, indicating excellent agreement.

As some of the McDonald’s omega coefficients were low, Spearman’ correlation was applied for global perception with the 16-P-CPQ, subscales and the 4-FIS in the total sample. The respective coefficients were statistically significant (*p* < 0.001) (Table 5), indicating a valid perception of the OHRQoL.

For building the logistic regression models, two dependent variables, the 16-P-CPQ total scale and the 4-FIS, were dichotomized by the median of the scores of the entire sample, and the threshold value was 6 and 3, respectively. Some variables were dichotomized to better suit the application of the regression model, such as ‘tooth brushing frequency’ and ‘teeth grinding’. The results of the univariate logistic regression are shown in Table 6, in which only the independent variables with *p* values equal to or less than 0.15 are indicated and entered into the multivariate logistic regression model.

The variables that remained statistically significant in the multivariate model (*p* < 0.05) are presented in Table 7. Thus, older ages (OR = 1.24), not brushing or brushing teeth once a day (OR = 2.21), teeth grinding (OR = 2.20) and gingival bleeding (OR = 3.34) were determined as having a greater chance of the parents perceiving a worse OHRQoL for their children and adolescents. The family functioning and activities due to oral conditions of the child was less affected in families with parents with an elementary school degree (OR = 0.314) and most affected in families whose incomes was less or equal to the minimum wage (OR = 3.049).

## 4. Discussion

The OHRQoL is a broad health concept which can be inserted into the holistic health model. It comprises the individual as a whole and defines how much the individual’s quality of life is affected by their oral health. In this context, this study aimed to investigate the parental perception of the OHRQoL in children and adolescents with ASD compared to unaffected individuals in a Brazilian population using an online design.

Parents/guardians’ perceptions of a children’s oral health may be influenced by their awareness of the important contribution oral health has to the child’s overall health [3]. In some instances, the relevance of the oral condition can be neglected because the parents of children with ASD have enormous burdens related to the child’s general health problems [30]. Other factors can also influence the parents’ perception of their child’s oral conditions, such as education level, family income, and occupational situation. If such factors are deficient, oral health care may be compromised [19].

Regarding the study participants, most of the questionnaires were answered by mothers, a fact also observed in previous studies [3,19,22,31] reaffirming the educational responsibility in health culturally designated to maternal figures, who assume the role of primary caregivers.

As expected, the predominance of male children and adolescents with ASD was observed in the present study. According to Centers for Disease Control and Prevention (CDC) reports, the prevalence of ASD in 4-year-old boys is 3.4 times higher than in girls [32], while at the age of eight, the respective value is 4.2 [33]. This difference can be attributed to missed clinical symptoms in girls, and subsequent gender diagnostic bias, as pointed out by Loomes et al. [34]; considering the male-to-female ratio of 3:1 reporting that girls who meet criteria for ASD were at a disproportionate risk of not receiving a clinical diagnosis.

In this study, most children and adolescents in the ASD group were diagnosed between the ages of two and six, reflecting the reality of late diagnosis, since ASD can be diagnosed as early as 18 months [35]. A recent systematic review analyzed the mean age at ASD diagnosis including 56 studies from 40 different countries and found a mean diagnosis of 60.48 months, ranging from 30.90 to 234.57 months [36]. Late diagnoses limit the opportunity for learning during the first years of life, a critical period in a child’s development. Thus, several global and regional efforts to enhance early detection, diagnosis and treatment of ASD must be strengthened, to ensure timely detection and management of ASD in primary care [36,37].

A high number of parents in the ASD group reported the use of medication, which may be related to the prevalence of a level 2 (moderate) diagnosis found in the present sample, who normally need substantial support from their relatives and exhibit more disruptive behaviors. Besides the ASD characteristics, comorbid conditions can also manifest, differing in symptomatology, prevalence, and treatability from children and adolescents with normal development [38]. Because the complex psychopathology, structured therapy, such as receiving ABA for the ASD group, plus parent training, cannot be enough, especially for children with intellectual disability (ID) and multiple comorbidities [38]. Thus, pharmacologic agents and individualized treatment are mandatory [39].

The minimum frequency of oral hygiene “No brushing/once a day” was proportionally higher in the ASD group, in addition to gingival bleeding, even if not statistically significant when compared to the UCA group. Such data suggest poor oral hygiene habits in the ASD group, which can be explained by the lack of manual skills, social interaction, and sensory processing disorders (difficulties with toothbrush, taste of toothpaste), characteristic of ASD [40]. Children and adolescents with ASD need support from their caregivers to perform brushing and sometimes refusal and non-cooperation make it difficult to maintain oral health, making them more susceptible to biofilm accumulation and consecutive caries lesions. Similar results were observed by Qiao et al. [24] who compared the oral health status of children with ASD and unaffected children, reporting that 99.2% of individuals with ASD suffer from at least one oral comorbidity, a higher prevalence of halitosis and poor oral habits. In a recent review, Gao and Liu [41] referred to effective methods for promoting daily oral health care in ASD children, such as visual education and social stories, ABA therapy, oral health education for guardians, interdisciplinary collaboration and professional level improvement. However, for children to become independent regarding oral hygiene, the degree of ASD plays an important role.

A high frequency of bottle-feeding in the ASD group was found. This association may have occurred due to the characteristics of ASD, as they may have sensory hypersensitivity or dislike changes, and many are unable to make the complete transition from bottle to other forms of feeding [42]. Thus, early diagnosis of atypical eating behaviors is important to intervene properly in time [42]. Furthermore, sugar consumption is an important variable due to its close relationship with dental caries. Although there was no significant difference in sugar consumption between the two groups, this fact is more worrying for the ASD group, in which tooth brushing was less frequent than the UCA group. Similar results were found by Moorthy et al. [43] who analyzed the sugar exposure of children aged 5–12 years with and without ASD using the Dental Diet Diary (D3) mobile app, with no difference between the groups. However, in the study by Moorthy et al. [43], oral hygiene practices were better in children with ASD, differing from the results of the present study, perhaps due to different forms, questions and how they were applied.

In addition, children of the ASD group showed statistically significant results for the variable ‘grinding teeth’, suggesting a greater propensity of children with ASD to present this condition. However, systematic reviews [9,10] have stated that this association has not yet been well established, and that many factors such as anxiety, stress and continuous medication use may contribute to neurological impulses and consequent involuntary muscular movements.

The present study revealed, through parental perception, a worse OHRQoL in children and adolescents of the ASD group when compared to the UCA group. This analysis was observed by comparing the final score data and the 16-P-CPQ subscales in the ASD and UCA groups, except for the “OS” subscale, which showed no statistical difference, as found previously [22]. This similarity between the groups can be explained by the fact that the “OS” subscale is the most objective, evaluating the symptoms and oral discomforts already present (pain, gingival bleeding, halitosis), of which the parents’ opinion does not directly influence the response. On the other hand, the well-being subscales (EWB and SWB) may have been influenced by the behavioral condition of the disorder in the responses by parents in the ASD group.

Significantly higher 4-FIS scores were also found in the ASD group, revealing a strong impact of the child’s or adolescent´s oral health condition on the change in family routine, such as interrupted sleep of the parents or even absence from work. Based on a similar analysis, Pani et al. [22] compared the family impact of the OHRQoL between families with and without children with autism, using the FIS as a questionnaire, and found no statistical difference; however, they observed significant differences in the parental emotion and financial components of the FIS scale.

The global perception questions (OH and OWB) analyze the criticality of parents regarding the oral health condition of their children. In the present study, more parents of the ASD group rated the oral health of their children as “fair/poor”, in line with the oral health outcomes found in the group, such as low brushing frequency, premature tooth loss and gingival bleeding. At the same time, the OWB responses showed the greatest concern for parents of the ASD group, as 24% of them considered that the oral health affected the quality of life “somewhat/a lot”.

The reliability test indicated substantial internal consistency for the 16-P-CPQ total; however, for the subscales the internal consistency ranged from weak to moderate. Nqcobo et al. [3] stated that a positive correlation between the global perception and the P-CPQ indicates that the OHRQoL perception scores can be considered valid. Thus, in the present study, the correlations were carried out and the respective coefficients were significant, indicating that the parents accurately perceived the OHRQoL of their children. The ICC test on a 20% sample indicated excellent agreement, with data ranging from 0.969 to 1 in the subscales. This result can be explained by the fact that the subscales in the short form of the questionnaire and FIS have four questions in each.

From the univariate logistic regression model, the presence of ASD was associated with a worse OHRQoL perception by the parents (OR = 2.17), suggesting that children with ASD were at least two times more likely to have a poorer OHRQoL. In fact, they are more prone to developing oral diseases [6,44] and poor oral health can cause pain, discomfort, and problems in social interactions that are easily perceived by caregivers. Age was another variable positively associated with OHRQoL, being a predictor of a worse parental perception of the OHRQoL (OR = 1.159). The parents’ perception worsened with the child’s age, as the severity of the oral diseases increased over time, agreeing with Qiao et al. [24]. In addition, oral hygiene habits were also associated with the 16-P-CPQ in the univariate logistic regression. When parents were responsible for brushing, their perceptions of their child’s OHRQoL were three times more likely to be worse. Likewise, the low frequency of brushing, that is, 0–1x/day, revealed parents were twice as likely of perceiving worse a QHRQoL. These findings can be attributed to the difficulties in controlling brushing behavior, as found by AlOtaibi et al. [45], demonstrating the importance of improving education and instructions in good oral health practices for this population and their caregivers, especially in ASD. Moreover, other sociodemographic and dental variables were considered eligible to enter into the multivariate logistic regression (*p* < 0.15). In the final model, the frequency of tooth brushing, gingival bleeding, and teeth grinding were significant predictors of worst parental perception of their children’s OHRQoL. The two dental variables, frequency of tooth brushing and gingival bleeding, supported the idea that poor oral hygiene is the main cause of the gingival diseases, as concluded by AlOtaibi et al. [45] in a recent systematic review.

Teeth grinding was negatively associated with OHRQoL (OR = 2.204). In other words, children with bruxism were two times more likely to have a poor OHRQoL reported by their parents. Conversely, Antunes et al. [46] found a low impact of bruxism on the OHRQoL using ECOHIS when compared to other studies with different comorbidities. Although ASD was not associated with the OHRQoL in a multivariate logistic regression, it is important to note that children with ASD are more likely to have sleep problems, contributing to stereotypic behavior, and involuntary movement of the masticatory muscles [10].

Regarding the dependent variable 4-FIS, in an univariate logistic regression there was no strong association with the variables studied, but those with *p* < 0.15 were inserted into the multivariate model. The final model showed that families with an income of less than or equal to the minimum wage were at least three times more affected by the oral health conditions of their children than families that received two to three times the minimum wages (OR = 3.049). This can be explained by the inequities in access to oral health information and difficulties in implementing these practices for these families, agreeing with Knorst et al. [47]. These authors [42] observed that a low socioeconomic status was associated with a worse OHRQoL in different age groups, and the impact of oral health conditions was 30% higher for families with a low socioeconomic status. Surprisingly, in this study, parents with an elementary school degree had a lower family impact compared to parents with a high school degree.

However, limitations of this study can be pointed out, such as the low response from the institutions to recruit participants, with a response rate of 31.5%, that limited the sample size, despite this the number of participants was sufficient to ensure good results. In addition, for the test–retest analysis, the application of the second form was hampered by the participants’ refusal to respond a second time, interfering with the reliability of the questionnaires, despite the high ICC values obtained and the positive correlations between the global perception and questionnaires. Moreover, the impact of well-being domains may have been affected in the ASD group due to the parents’ difficulty in interpreting the information about their children’s emotional and social interactions, generating many ‘0’ scores, making it difficult to perform certain statistical tests. However, these aspects did not affect the expected scope of the study. On the other hand, the design of the online forms did not allow for missing data, ensuring confident results, despite the above comments about the well-being domains. Furthermore, most of the respondents were mothers who are normally the primary caregivers, therefore giving reliable answers. In addition, the clinical oral examination of children and adolescents could not be performed due to the restrictions imposed by the COVID-19 pandemic, requiring further studies with the respective assessments in order to observe specific oral conditions more accurately.

## 5. Conclusions

The parents of children and adolescents with ASD had a worse perception related to the QHRQoL when compared to the unaffected population. ‘Frequency of tooth brushing’, ‘gingival bleeding’, and ‘teeth grinding’ were predictors of a worst parental perception of their children’s OHRQoL. Families with low socioeconomic conditions, such as families with an income equal to or less than the minimum wage, were more strongly affected by the oral conditions of their children.

Oral health professionals must be prepared to offer specialized clinical care to patients with ASD, as well as be able to guide parents regarding oral hygiene habits, reducing sugar in the diet, modifying harmful oral habits and highlighting the importance of oral health to improve both the general health and the patient’s quality of life.

## Figures and Tables

**Figure 1 ijerph-20-01151-f001:**
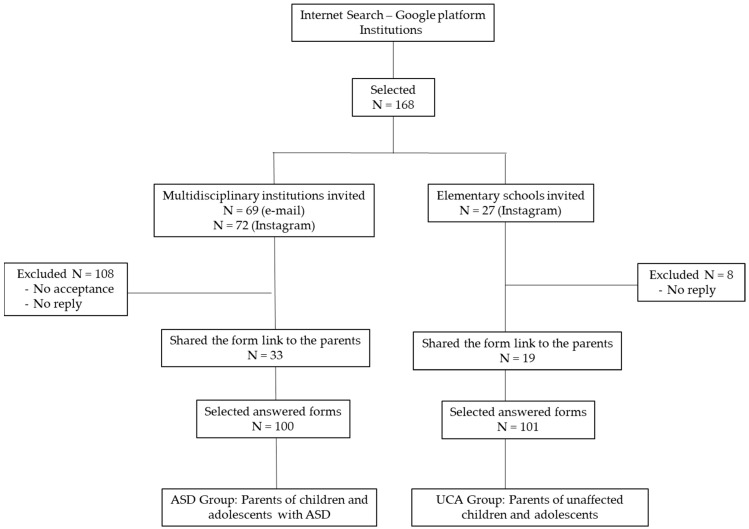
Flowchart referring to the invitation and acceptance of institutions to obtain the sample.

**Table 1 ijerph-20-01151-t001:** Distribution of sociodemographic variables, oral habits, and dental care for both groups.

		ASD Group (n = 100)	UCA Group (n = 101)	*p* Value
Age	Mean (SD)	8.38 (2.38)	9.11 (1.96)	Mann–Whitney *p* = 0.019
	Median (25th; 75th)	8 (6.0; 10.0)	9 (8.0; 11.0)	
	Range	6–14	6–14	
Sociodemographic [n (%)]				
Sex	Male	65 (65)	55 (54.5)	χ^2^ *p* = 0.128
	Female	35 (35)	46 (45.5)	
ASD diagnosis (age)	<2 years	36 (36)	-	χ^2^ goodness of fit *p* < 0.001
	2–6 Years	54 (54)	-	
	>6 years	10 (10)	-	
ASD levels	Level 1	37 (37)	-	χ^2^ goodness of fit *p* < 0.001
	Level 2	51 (51)	-	
	Level 3	12 (12)	-	
Medication use	Yes	53 (53)	6 (5.90)	χ^2^ *p* < 0.01
	No	47 (47)	95 (94.1)	
Parent’s education	Elementary School	24 (24)	15 (14.9)	χ^2^ *p* < 0.01
	High school	16 (16)	55 (53.5)	
	College	44 (44)	26 (25.8)	
	Postgraduate studies	16 (16)	5 (5)	
Employment status	Working	79 (79)	82 (81.2)	χ^2^ *p* = 0.698
	Unemployed	21 (21)	19 (18.8)	
Family income	≤1 minimum wage	17 (17)	8 (7.9)	χ^2^ *p* < 0.001
	2–3 minimum wage	34 (34)	67 (66.3)	
	≥4 minimum wage	49 (49)	26 (25.7)	
Respondent	Mother	90 (90)	94 (93.1%)	Fisher’s exact test *p* = 0.062 *p* < 0.001 *
	Father	10 (10)	4 (4.0%)	
	Other	0	3 (3.0%)	
Feeding [n (%)]				
Mealtime	Yes	81 (81)	76 (75.2)	χ^2^ *p* = 0.324
	No	19 (19)	25 (24.8)	
Bottle feeding	Don’t use	66 (66)	94 (93)	Fisher’s exact test *p* < 0.001
	Once a day	7 (7)	2 (2)	
	Twice a day	22 (22)	5 (5)	
	Three times a day	5 (5)	0	
Sugar consumption	Don’t consume	0	0	χ^2^ *p* = 0.493
	Once a day	46 (46)	44 (43.6)	
	2–3 times a day	36 (36)	32 (31.7)	
	More than 3 times a day	18 (18)	25 (24.8)	
Oral habits and dental care [n (%)]				
Tooth brushing frequency	Don’t brush	8 (8)	0	Fisher’s exact test *p* < 0.001
	Once a day	42 (42)	24 (23.8)	
	Twice a day	23 (23)	44 (43.6)	
	Three times a day	27 (27)	33 (32.7)	
Responsible for tooth brushing	Parents	75 (75)	7 (6.9)	χ^2^ *p* < 0.001
	Child	3 (3)	33 (32.7)	
	Both	22 (22)	61 (60.4)	
Dentist appointment	Yes	62 (62)	84 (83.2)	χ^2^ *p* < 0.001
	No	38 (38)	17 (16.8)	
Premature tooth loss	Yes	40 (40)	8 (7.9)	χ^2^ *p* < 0.001
	No	60 (60)	93 (92.1)	
Gingival bleeding	Yes	12 (12)	12 (11.9)	χ^2^ *p* = 0.979
	No	88 (88)	89 (88.1)	
Dental Trauma	Yes	32 (32)	33 (32.7)	χ^2^ *p* = 0.919
	No	68 (68)	68 (67.3)	
Deleterious oral habits	No	70 (70)	86 (85.1)	Fisher’s exact test *p* = 0.004
	Digital suction	2 (2)	3 (3)	
	Pacifier	1 (1)	1 (1)	
	Feeding bottle	20 (20)	6 (5.9)	
	Onychophagia	7 (7)	5 (5)	
Mouth breathing	Yes	56 (56)	40 (39.6)	χ^2^ *p* = 0.020
	No	44 (44)	61 (60.4)	
Teeth grinding	Never	52 (52)	73 (72.3)	χ^2^ *p* = 0.005
	Sometimes	39 (39)	26 (25.7)	
	Many times	9 (9)	2 (2)	

ASD, autism spectrum disorder; UCA, unaffected children and adolescents; * χ^2^ goodness of fit.

**Table 2 ijerph-20-01151-t002:** Descriptive data of the questionnaires (16-P-CPQ total, subscales, global and 4-FIS).

		ASD Group (N = 100)	UCA Group (N = 101)	*p* Values ^1^	Effect Size ^2^
16-P-CPQ total scale	Mean (SD)	8.51 (5.84)	6.12 (4.58)	*p* = 0.002	0.254
	Median (25th; 75th)	6.5 (4; 11)	5 (4; 8)		
	Range	0–23	1–23		
Subscales					
OS	Mean (SD)	3.58 (2.69)	3.03 (2.61)	*p* = 0.096	0.134
	Median (25th; 75th)	4.00 (2.00; 5.00)	2.00 (1.00–5.00)		
	Range	0–11	0–11		
FL	Mean (SD)	3.27 (2.44)	2.34 (1.94)	*p* = 0.007	0.215
	Median (25th; 75th)	3.00 (2.00; 5.00)	2.00 (1.00; 3.00)		
	Range	0–9	0–9		
EWB	Mean (SD)	0.95 (1.32)	0.49 (0.88)	*p* = 0.021	0.157
	Median (25th; 75th)	0 (0; 2)	0 (0; 1)		
	Range	0–5	0–3		
SWB	Mean (SD)	0.71 (1.13)	0.27 (0.65)	*p* = 0.002	0.195
	Median (25th; 75th)	0 (0; 1)	0 (0; 0)		
	Range	0–5	0–3		
Global Perception	Mean (SD)	2.78 (1.85)	1.74 (1.29)	*p* < 0.001	0.316
	Median (25th; 75th)	3.00 (1.00; 4.00)	2.00 (1.00; 2.00)		
	Range	0–6	0–6		
4-FIS	Mean (SD)	4.08 (3.21)	2.95 (2.52)	*p* = 0.025	0.182
	Median (25th; 75th)	3 (2.00; 6.25)	3 (1.00; 4.00)		
	Range	0–11	0–11		

P-CPQ: parental-caregiver perception questionnaire; FIS: family impact scale; ASD: autism spectrum disorder; UCA: unaffected children and adolescents; OS: oral symptoms; FL, functional limitations; EWB: emotional well-being; SWB: social well-being; ^1^ Mann–Whitney U, ^2^ rank biserial correlation.

**Table 3 ijerph-20-01151-t003:** Distribution of responses to the global perception questions.

	ASD Group N (%)	UCA Group N (%)	
Global rating of oral health			
Excellent	12 (12.0)	14 (13.86)	Fisher’s exact test *p* < 0.001
Very good	26 (26.0)	51 (50.50)	
Good	24 (24.0)	30 (29.70)	
Fair	32 (32.0)	3 (2.97)	
Poor	6 (6.0)	3 (2.97)	
Global rating of overall well-being (affected)			
Not at all	47 (47.0)	75 (74.36)	χ^2^ *p* < 0.001 *
Very little	29 (29.0)	15 (14.85)	
Some	17 (17.0)	4 (1.98)	
A lot	7 (7.0)	7 (6.93)	
Very much	0	0	

* chi-square test; ASD, autism spectrum disorder; UCA, unaffected children and adolescents.

**Table 4 ijerph-20-01151-t004:** Scale reliability statistics for the 16-P-CPQ total and subscales, for the 4-FIS and global perception.

	ASD Group		UCA Group	
	McDonald’s ω	ICC (95% CI) (n = 10)	McDonald’s ω	ICC (95% CI) (n = 10)
16-P-CPQ total	0.778	0.982 (0.932–0.995)	0.773	0.996 (0.985–0.999)
Subscales				
OS	0.684	0.995 (0.980–0.999)	0.692	1
FL	0.638	0.969 (0.887–0.992)	0.502	1
EWB	0.471		0.444	
SWB	0.380		0.469	
Global perception	0.708	1	0.227	0.974 (0.906–0.993)
4-FIS	0.743	0.982 (0.932–0.995)	0.715	0.935 (0.776–0.983)

P-CPQ, parental–caregiver perception questionnaire; FIS, family impact scale; ASD, autism spectrum disorder; UCA, unaffected children and adolescents; OS: oral symptoms; FL, functional limitations; EWB: emotional well-being; SWB: social well-being.

**Table 5 ijerph-20-01151-t005:** Correlation between global perception questions and the 16-P-CPQ total, subscales and 4-FIS.

	Global Oral Health (n = 201)		Global Well-Being Teeth (n = 201)	
	R *	*p* Value	R	*p* Value
16-P-CPQ total	0.421	<0.001	0.662	<0.001
Subscales				
OS	0.301	<0.001	0.366	<0.001
FL	0.393	<0.001	0.692	<0.001
EWB	0.295	<0.001	0.431	<0.001
SWB	0.214	0.002	0.356	<0.001
4-FIS	0.339	<0.001	0.578	<0.001

* Spearman’s rho; P-CPQ, parental–caregiver perception questionnaire; FIS, family impact scale; OS: oral symptoms; FL, functional limitations; EWB: emotional well-being; SWB: social well-being.

**Table 6 ijerph-20-01151-t006:** Univariate logistic regression. The independent variables only with *p* values equal to or less than 0.15.

	Predictor	Estimate	Odds Ratio	*p* Value	95% CI	R^2^_N_
**Dependent variable: 16-P-CPQ total dichotomized (Median 6)**						
Groups	ASD—UCA	0.775	2.17	0.008	1.22–3.86	0.047
	Age	0.148	1.159	0.026	1.02–1.32	0.034
Employment status	Unemployed—Working	0.521	1.683	0.143	0.84–3.38	0.014
Responsible for tooth brushing	Parent—child/adolescent	1.001	2.721	0.024	1.14–6.50	0.037
	Both—child/adolescent	0.633	1.882	0.156	0.79–4.51	
Mealtime	No–Yes	0.653	1.921	0.058	0.98–3.77	0.024
Dentist appointment	No–Yes	0.526	1.692	0.100	0.90–3.17	0.018
Dental Trauma	Yes–No	0.575	1.778	0.060	0.98–3.24	0.024
Teeth grinding	Yes–No	0.505	1.657	0.088	0.93–2.96	0.019
Gingival bleeding	No–Yes	−0.663	0.515	0.130	0.22–1.22	0.053
Tooth brushing frequency	0–1x/day—2–3x/day	0.849	2.337	0.005	1.30–4.21	0.053
**Dependent variable: 4-FIS dichotomized (Median 3)**						
Parent´s Education	Elementary School–High School	−0.665	0.514	0.109	0.23–1.16	0.020
	College–High School	−0.029	0.971	0.931	0.50–1.88	
	Postgraduate–High School	−0.0671	0.935	0.893	0.35–2.48	
Family income	≤1 minimum wage—2–3 minimum wages	0.705	2.023	0.122	0.83–4.94	0.016
	≥4 minimum wage—2–3 minimum wages	0.112	1.119	0.715	0.61–2.04	
Tooth brushing frequency	0–1x/day—2–3x/day	0.442	1.556	0.133	0.87–2.77	0.015

P-CPQ, parental–caregiver perception questionnaire; FIS, family impact scale; ASD, autism spectrum disorder; UCA, unaffected children and adolescents; CI, confidence interval; R^2^_N_, Nagelkerke’s R^2^; Note. Estimates represent the log odds of “16-P-CPQ dichotomized = 1” (median > 6) vs. of “16-P-CPQ dichotomized = 0” (median < 6). VIF, Variance Inflation Factor = 1 and Tolerance = 1 for all univariate regressions.

**Table 7 ijerph-20-01151-t007:** Multivariate logistic regression. The independent variables only associated with the 16-P-CPQ and 4-FIS dichotomized scores (*p* < 0.05).

Predictor (Independents)	Estimate	Odds Ratio	*p*-Value	95% CI	Collinearity	
Dependent variable: 16-P-CPQ dichotomized; Overall Model Test: χ^2^ = 35.9, *p* < 0.001. R^2^_N_ = 0.221						
Intercept	−3.233	0.040	<0.001	0.008–0.193	VIF	Tolerance
Age	0.213	1.237	0.007	1.06–1.44	1.10	0.908
Tooth brushing frequency						
0–1x/day—2–3x/day	0.793	2.210	0.025	1.10–4.42	1.10	0.908
Teeth grinding						
Yes—No	0.790	2.204	0.028	1.09–4.46	1.12	0.894
Gingival bleeding						
Yes—No	1.206	3.341	0.030	1.12–9.96	1.18	0.851
Dependent variable: 4-FIS dichotomized; Overall Model Test: χ^2^ = 11.4, *p* = 0.076, R^2^_N_ = 0.074						
Parent’s education						
Elementary School—High School	−1.158	0.314	0.014	0.13–0.79	1.14	0.876
Family income						
≤1 minimum wage— 2–3 minimum wage	1.115	3.049	0.029	1.12–8.28	1.2	0.830

Note: 1. P-CPQ, parental–caregiver perception questionnaire; FIS, family impact scale; CI, confidence interval; R^2^_N_, Nagelkerke’s R^2^; 2. Estimates represent the log odds of “4-FIS dicot = 1” vs. “4-FIS dicot = 0”.

## Data Availability

The individual data presented in this study are available on request from the corresponding author. The data are not publicly available due to the privacy of institutions and participants.

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
