# Peer review of "Parental Perception of the Oral Health-Related Quality of Life of Children and Adolescents with Autism Spectrum Disorder (ASD)"

_ijerph, 2023, doi:10.3390/ijerph20021151_

Round 1

Reviewer 1 Report

The paper reports an original study, aiming to evaluate the parental perception of OHRQoL of the children and adolescents affected by autism spectrum disorder.

The study has been accurately planned and performed, collecting a significant quantity of data. The data are clearly reported, and the paper is well written.

I would have only one remark

Line 290

It comprises the child..” it would be better to write “It comprises the patient (or person).. “ since OHRQoL is related to every age, not only to children.

Author Response

We thank the Reviewer for kind comments on the manuscript.

Line 290

Reviewer: “It comprises the child..” it would be better to write “It comprises the patient (or person).. “ since OHRQoL is related to every age, not only to children.

Answer: We agree with the suggestion. The correction was made accordingly.

Reviewer 2 Report

Dear authors

Thank you for this nice reading. Overall, this work presents important information, and from the clinical point of view is very interesting to know the parents’ perception and attitude however, how can the pediatric dentist apply in in the consult. 

Below you will find my comments/concerns in detail regarding what needs further attention. 

Introduction:

Line 49 – 51 – “Some tools were developed over the years to assess the impact of oral health on chil- 49 dren's and adolescents’ quality of life, including the Child Perceptions Questionnaires (CPQ8- 50 10 and CPQ11-14) answered by children [7,9]”

You refer to the CPQ questionnaire, aimed at children and adolescents, why this one, are there others?

Line 65 - “Some studies have revealed a negative 65 impact of high caries…”

You could summarize the most common oral consequences in ASD children, in order to highlight that it is not just caries, poor oral hygiene and non-cooperation. I suggest reading 1 article on e.g. Malocclusion in ASD Children

Autism Spectrum Disorders and Malocclusions: Systematic Review and Meta-Analyses, 2022, Journal of Clinical Medicine (DOI: 10.3390/jcm11102727)

Materials and Methods

 Line 99 – “These forms 98 contained information about the study, the TCLE to indicate consent to participate…”

What does it mean TCLE

Line 117 - “in the third, two OHRQoL 116 questionnaires were applied, the Brazilian Portuguese short versions of the Parental-Care- 117 giver Perceptions Questionnaire (P-CPQ)...”

Is the nomenclature short version or Short form?

Line 123 – “Figure 1. Flowchart referring to the invitation and acceptance of institutions to obtain the sample”

Will this figure not be more appropriate in the results?

Results

Page 6 – Table 1 – Between “Responsible for tooth brushing” and “Dentist appointment 

“there is missing a line or it is a problem of formatting.

Discussion

Line 310-314 – “Late diagnoses 310 limit the opportunity for effective treatment during the first years of life...”

What do you consider effective treatment and and what specifically could be done to improve the quality of life

Conclusions

This conclusion was expected, so what can oral health professionals’ advice the parents regarding the ASD children

Reviewer 3 Report

Suggestion for Authors:

1. Starting from the Title of the article,

"Parent's perception about the oral health-related quality of life of children and adolescents with autism spectrum disorder (ASD) and reflecting in Chapter 4 and Chapter 5 can be improved.

1a. Chapter 4. Discussions can deal with the following issues:

-How are oral cavity problems affected and influenced by the erroneous perception of parents or official guardians about the oral health of children with ASD?

-Is it possible that through the ABA Therapy you mention at 320-321, children with ASD can become independent in brushing their teeth, copying an automaticity learned through Behavior Contracts or Video Modeling?

1b.Chapter 5 Conclusions may contain the personal opinion of the Authors about the topic and their proposal in order to improve the oral health of children with ASD (through programs offered to both children and parents).

2. OHRQOL is associated with:

 -Functional factors (mastication and speech)

-Psychological factors (appearance and self esteem)

- Social factors (intimacy and communication)

- Experience of pain or discomfort (acute and chronic)

2a.How do you explain the use and accuracy of the method (OHRQOL) of quantifying the oral health status of children with ASD by questioning parents or official guardians?

2b.Are they a mirror of the well-being of children with problems on the autistic spectrum, or is it necessary to implement a continuous screening program for these children in terms of oral health?

3.Chapter 5 Conclusions:

3a. Can the quality of life for children with ASD be improved by educating parents and official guardians regarding oral health and its maintenance?

Author Response

Reviewer 3

We thank the Reviewer for the questions and comments that can improved the manuscript.

  1. Starting from the Title of the article,

"Parent's perception about the oral health-related quality of life of children and adolescents with autism spectrum disorder (ASD) and reflecting in Chapter 4 and Chapter 5 can be improved.

1a. Chapter 4. Discussions can deal with the following issues:

-How are oral cavity problems affected and influenced by the erroneous perception of parents or official guardians about the oral health of children with ASD?

Answer: We have tried to adequately answer your question. We hope you agree.

Parents/guardians' perception of children's oral health may be influenced by awareness of the importance of oral health's contribution to overall health (Nqcobo et al., 2019). In some instances, the relevance of oral conditions can be neglected because parents of children with ASD can have enormous burdens related to the general health problems (Lewis et al. 2015). Other factors can also influence the parents' perception of their children's oral conditions, such as education level, family income, and occupational situation. If such factors are deficient, oral health care may be compromised (Puthiyapurayil et al 2022).

-Is it possible that through the ABA Therapy you mention at 320-321, children with ASD can become independent in brushing their teeth, copying an automaticity learned through Behavior Contracts or Video Modeling?

Answer: Yes, it is possible to improve the brushing, but we believed that the children become independent the ASD degree must be a role. We added some consideration about this aspect, as follows:

In a recent review, Gao and Liu (2022) referred to methods for promoting daily oral health care in ASD children, such as visual education and social stories, ABA, oral health education for guardians, interdisciplinary collaboration and professional level improvement. Nevertheless, for children to become independent regarding oral hygiene, the degree of ASD plays an important role.

1b.Chapter 5 Conclusions may contain the personal opinion of the Authors about the topic and their proposal in order to improve the oral health of children with ASD (through programs offered to both children and parents).

  1. OHRQOL is associated with:

 -Functional factors (mastication and speech)

-Psychological factors (appearance and self esteem)

- Social factors (intimacy and communication)

- Experience of pain or discomfort (acute and chronic)

2a.How do you explain the use and accuracy of the method (OHRQOL) of quantifying the oral health status of children with ASD by questioning parents or official guardians?

In our opinion, the OHRQoL cannot quantify the oral health status of children in general but it can accurately measure the impact of existing oral conditions on quality of life, since the respective instruments have been validated in many studies and parents serving as proxies for young children and those with special needs. Moreover, oral health in the present study was quantified with questions addressed in the second section of the google form, with selected variables that could be precisely evaluated by the parents/guardian. For example, dental caries was not included, since a precise diagnosis by trained and calibrated examiners is necessary.

2b .Are they a mirror of the well-being of children with problems on the autistic spectrum, or is it necessary to implement a continuous screening program for these children in terms of oral health?

As commented above, methods for promoting oral health care in ASD children include health education for guardians, interdisciplinary collaboration, and professional level improvement, thus a continuous screening program should be implemented.

3.Chapter 5 Conclusions:

3a. Can the quality of life for children with ASD be improved by educating parents and official guardians regarding oral health and its maintenance?

Yes, the quality of life for children with ASD can be improved by increasing oral health awareness, which requires parents’ education, highlighting the importance of oral health for general health, as added in the conclusion.

Round 2

Reviewer 2 Report

Dear authors

Thank you once again for this nice reading. Best of luck on upcoming projects and keep up the good work.

Kind regards

LBL

Reviewer 3 Report

Dear authors,

the Manuscript was improved well.

All the best in the future research